# Evaluation of clonal hematopoiesis and mosaic loss of Y chromosome in cardiovascular risk: An analysis in prospective studies

Sami Fawaz[1†], Severine Marti[2†], Melody Dufossee[3†], Yann Pucheu[1], Astrid Gaufroy[1], Jean Broitman[1], Audrey Bidet[2], Aicha Soumare[4], Gaëlle Munsch[4], Christophe Tzourio[4], Stephanie Debette[4], David-Alexandre Trégouët[4], Chloe James[2,3], Olivier Mansier[2,3*‡], Thierry Couffinhal[1,3*‡]

[1]CHU de Bordeaux, Service des Maladies Coronaires et Vasculaires, Pessac, France; [2]CHU de Bordeaux, Laboratoire d'hematologie, Pessac, France; [3]Univ. Bordeaux, INSERM, Biologie des maladies cardiovasculaires, Pessac, France; [4]Univ. Bordeaux, Bordeaux Population Health Research Center, INSERM, Bordeaux, France

**\*For correspondence:**
olivier.mansier@inserm.fr (OM);
thierry.couffinhal@inserm.fr (TC)

[†]These authors contributed equally to this work
[‡]These authors also contributed equally to this work

**Competing interest:** The authors declare that no competing interests exist.

## eLife assessment

In this small study involving patients with a history of myocardial infarction, Fawaz et al. found no significant contribution of clonal hematopoiesis and mosaic loss of the Y chromosome to the incidence of myocardial infarction and atherosclerosis. Although the evidence provided by the study is **incomplete** due to its small sample size, the findings are **valuable** for guiding future larger studies that will further investigate this significant and controversial subject.

## Abstract

**Background:** Clonal hematopoiesis of indeterminate potential (CHIP) was initially linked to a twofold increase in atherothrombotic events. However, recent investigations have revealed a more nuanced picture, suggesting that CHIP may confer only a modest rise in myocardial infarction (MI) risk. This observed lower risk might be influenced by yet unidentified factors that modulate the pathological effects of CHIP. Mosaic loss of the Y chromosome (mLOY), a common marker of clonal hematopoiesis in men, has emerged as a potential candidate for modulating cardiovascular risk associated with CHIP. In this study, we aimed to ascertain the risk linked to each somatic mutation or mLOY and explore whether mLOY could exert an influence on the cardiovascular risk associated with CHIP.

**Methods:** We conducted an examination for the presence of CHIP and mLOY using targeted high-throughput sequencing and digital PCR in a cohort of 446 individuals. Among them, 149 patients from the CHAth study had experienced a first MI at the time of inclusion (MI(+) subjects), while 297 individuals from the Three-City cohort had no history of cardiovascular events (CVE) at the time of inclusion (MI(-) subjects). All subjects underwent thorough cardiovascular phenotyping, including a direct assessment of atherosclerotic burden. Our investigation aimed to determine whether mLOY could modulate inflammation, atherosclerosis burden, and atherothrombotic risk associated with CHIP.

**Results:** CHIP and mLOY were detected with a substantial prevalence (45.1% and 37.7%, respectively), and their occurrence was similar between MI(+) and MI(-) subjects. Notably, nearly 40% of CHIP(+) male subjects also exhibited mLOY. Interestingly, neither CHIP nor mLOY independently resulted in significant increases in plasma hs-CRP levels, atherosclerotic burden, or MI incidence.

Moreover, mLOY did not amplify or diminish inflammation, atherosclerosis, or MI incidence among CHIP(+) male subjects. Conversely, in MI(-) male subjects, CHIP heightened the risk of MI over a 5 y period, particularly in those lacking mLOY.

**Conclusions:** Our study highlights the high prevalence of CHIP and mLOY in elderly individuals. Importantly, our results demonstrate that neither CHIP nor mLOY in isolation substantially contributes to inflammation, atherosclerosis, or MI incidence. Furthermore, we find that mLOY does not exert a significant influence on the modulation of inflammation, atherosclerosis burden, or atherothrombotic risk associated with CHIP. However, CHIP may accelerate the occurrence of MI, especially when unaccompanied by mLOY. These findings underscore the complexity of the interplay between CHIP, mLOY, and cardiovascular risk, suggesting that large-scale studies with thousands more patients may be necessary to elucidate subtle correlations.

**Funding:** This study was supported by the Fondation Cœur & Recherche (the Société Française de Cardiologie), the Fédération Française de Cardiologie, ERA-CVD (« CHEMICAL » consortium, JTC 2019) and the Fondation Université de Bordeaux. The laboratory of Hematology of the University Hospital of Bordeaux benefitted of a convention with the Nouvelle Aquitaine Region (2018-1R30113-8473520) for the acquisition of the Nextseq 550Dx sequencer used in this study.

**Clinical trial number:** NCT04581057.

## Introduction

Atherothrombosis is the main cause of death worldwide. Traditional cardiovascular risk factors (CVRF), such as diabetes, smoking, dyslipidemia, and hypertension, explain 70 to 75% of cardiovascular events suffered by patients. However, a significant part of these events remains unexplained given that 25 to 30% of people without any evident cause can present an atherosclerotic cardiovascular event (CVE), whereas not all high-risk subjects (according to traditional CVFR) experience such an event (*Berry et al., 2012*).

Recently, CHIP has emerged as a potential new risk factor for cardiovascular diseases (*Jaiswal et al., 2014*). This condition results from the acquisition by a hematopoietic stem cell of somatic mutations in leukemia-driver genes, leading to the clonal expansion of a population of hematopoietic cells without any clinical or biological sign of hematological malignancy. The definition of CHIP proposed to date, requires the detection of the mutation at a variant allele frequency (VAF) of more than 2%, representing a proportion of mutated cells of more than 4% (*Steensma et al., 2015*). The most commonly mutated genes in CHIP are *DNA Methyltransferase 3* A (*DNMT3A*) and *Ten Eleven Translocation 2* (*TET2*). In 2014, Jaiswal et al showed that CHIP was associated with a decreased survival mainly because of an increased atherothrombotic mortality. In particular, they observed a 2.0-fold increased risk of MI and a 2.6-fold increased risk of ischemic stroke (*Jaiswal et al., 2014*). In 2017, these data were confirmed, showing a 1.9-fold increased risk of coronary heart disease in the presence of CHIP, independently of traditional CVRF (*Jaiswal et al., 2017*). At the same time, a causative role of CHIP in inducing atherosclerosis has been demonstrated in animal models through the induction of a proinflammatory state (*Fuster et al., 2017*; *Jaiswal et al., 2017*). More recently, *Kessler et al., 2022* showed in 454,803 subjects from the UK Biobank that the association of CHIP with atherothrombotic events was restricted to high VAF clones (i.e. ≥10%) with a much lower risk than initially demonstrated (HR=1.11). But even with these criteria, no association between CHIP and atherothrombotic events was found in a validation cohort of 173 585 subjects (*Kessler et al., 2022*), which has also been suggested in another work studying several hundred thousand subjects (*Kar et al., 2022*). Thus, the impact of CHIP on atherothrombosis in humans is not totally evident, possibly because of the existence of yet unidentified modulating factors that could potentiate or counteract the effect of CHIP. For example, the p.Asp358Ala variant of the IL6 receptor gene has been shown to decrease the atherothrombotic risk associated with CHIP (*Bick et al., 2020*; *Vlasschaert et al., 2023a*). However, the impact of other genetic variants on the cardiovascular risk associated with CHIP remains unknown.

Gain or loss of chromosomes in hematopoietic cells appears to be as frequent as the acquisition of somatic mutations during aging (*Saiki et al., 2021*). In particular, mLOY has been shown to be frequent in male subjects without evidence of hematological malignancy (*Wright et al., 2017*; *Zhou et al., 2016*). mLOY was associated with cardiovascular diseases (*Loftfield et al., 2018*; *Sano et al., 2022*), and can be detected in a rather high proportion of subjects with CHIP (*Ljungström et al., 2022*;

*Zink et al., 2017*). Finally, while *TET2* mutations promote inflammation and atherosclerosis in mouse models (*Jaiswal et al., 2017*), mLOY was shown to switch macrophages from a pro-inflammatory to a pro-fibrotic phenotype (*Sano et al., 2022*). Thus, because of their opposite effect on macrophages phenotype, mLOY could balance the effect of CHIP regarding the induction of inflammation and thus decrease the development of atherosclerosis and the resulting atherothrombotic risk. We thus hypothesized that mLOY could modulate the effect of CHIP in inducing inflammation, atherosclerosis and triggering atherothrombotic events.

In this study, we used sensitive technics to determine the prevalence of both CHIP and mLOY in two cohorts of subjects. We sought to determine whether CHIP and mLOY significantly increase the cardiovascular risk separately. We also searched to determine in humans whether mLOY could impact the effect of CHIP on inflammation, atherosclerosis burden or atherothrombotic risk.

# Methods

**Key resources table**

| Reagent type (species) or resource | Designation | Source or reference | Identifiers | Additional information |
|---|---|---|---|---|
| Sequence-based reagent | Primer-amel-Fwd | This paper | PCR primers | CCCCTGGGCACTGTAAAGAAT |
| Sequence-based reagent | Primer-amel-Rev | This paper | PCR primers | CCAAGCATCAGAGCTTAAACTG |
| Sequence-based reagent | Probe-amelX | This paper | PCR probe | CCAAATAAAGTGGTTTCTCAAGT |
| Sequence-based reagent | Probe-amelY | This paper | PCR probe | CTTGAGAAACATCTGGGATAAAG |
| Commercial assay or kit | ddPCR supermix for Probes (no dUTP) | Biorad | PCR mix | |
| Commercial assay or kit | SureSelect XT Low Input kit | Agilent | | NGS custom RNA-baits panel |
| Software, algorithm | Quantasoft | Biorad | | Analysis software |
| Software, algorithm | R | CRAN | https://www.r-project.org/ | |

## Patients

For this study, we recruited 446 patients: 149 with a first MI and 297 without a MI or other CVE at inclusion. The 149 subjects with a MI were enrolled in the CHAth study between March 2019 and October 2021 (MI(+) subjects). They were included 4+/-2 mo after the acute event, in order to assess their basal inflammatory state. In the presence of any clinical sign or factor associated with inflammation, the appointment was reported in order not to skew biological and genetic data. Additionally, we ensured that the subjects had not been vaccinated against SARS-Cov2 within 15 d of enrollment. The study was approved by the institutional review board (IRDCB 2019-A02902-05), and registered (https://www.clinicaltrials.gov: NCT04581057). All participants gave written informed consent before inclusion in the study.

As a control cohort, we selected subjects without any history of CVE at inclusion in the Three-City (3 C) study cohort (MI(-) subjects) (*3C Study Group, 2003*). The 3 C study is a prospective study that enrolled 9294 subjects of 65 y or more who were selected upon electoral lists. These subjects were followed for several years (up to 12 y) to detect the development of dementia from a vascular origin. As such, they benefitted from a stringent cardiovascular follow-up, with adjudication of all cardiovascular events (in particular occurrence of MI). Among these subjects, we selected the 297 subjects who did not present any CVE before inclusion. Seventy-nine of them presented a MI during follow-up. The remaining 218 subjects had no atherothrombotic event during follow-up, and were matched on age, sex, and CVRF with those who had a MI during follow-up.

## Objectives and methodology to determine the number of subjects to include in the CHAth study

The CHAth study was an observational transversal monocentric study. It aimed to evaluate the prevalence of CHIP in patients over 75 presenting with a first CVE and to determine if CHIPs are more frequent in this population compared to a control cohort without CVE (recruited from the Three-City

study cohort or 3 C). To determine the number of patients necessary to achieve our objectives, we considered a CHIP prevalence of 20% in the general population after the age of 75 y, as estimated by *Genovese et al., 2014*, and *Jaiswal et al., 2017*; *Jaiswal et al., 2014*. At this time the relative risk of MI associated with CHIP was shown to be 1.7, leading to an expected prevalence of CHIP of 37% in subjects who presented a MI. Based on these hypotheses, the recruitment of 112 patients in the CHAth was estimated to be sufficient to show a higher prevalence of CHIP in MI(+) patients compared to MI(-) subjects with a statistical power of 0.90 at a type I error rate of 5%. Our study was not designed to show an effect of CHIP on incident MI during follow-up, including in the 297 MI(-) subjects from the 3 C study who were used as control subjects for MI(+) patients.

## Patients' inclusion in the CHAth Study

Eligible patients were ≥75 years of age and admitted for their first acute coronary event, without any history of other previous cardiovascular events. According to guidelines, a myocardial infarction was defined as a myocardial injury with clinical evidence of myocardial ischemia, with a significant change in cTroponin levels above the 99th percentile and at least one of the following criteria: symptoms of myocardial ischemia, ischemic ECG changes, development of pathological Q waves, imaging evidence of loss of viable myocardium or new regional wall motion abnormality, identification of a coronary thrombus by angiography (*Thygesen et al., 2018*). Patients with type 1 myocardial infarctions, and type 2 if there was evidence of atherosclerotic coronary disease, were included. Those with myocardial infarctions due to vasospasm, microvascular dysfunction, non-atherosclerotic coronary dissection, or oxygen supply/demand imbalance alone were excluded.

The other inclusion criterion was the absence of hematological malignancy, known or revealed by blood count at admission at the time of the event. If an abnormality was detected in the blood count at the time of admission, a complementary analysis was performed to search for a non-neoplastic cause (hemoconcentration, dilution, inflammation due to an active and reversible condition, vitamin or iron deficiency, bleeding not related to an active neoplasia). If no reversible cause was found and/or if the blood count was not normalized during the hospital stay, the patient was excluded from the study. Hemoglobin was considered normal if between 12 g/dL and 16 g/dL in women, and 13 g/dL and 17 g/dL in men; platelets count had to be between 150 and 400 G/L, and white blood cells were expected to be between 4 and 10 G/L.

Exclusion criteria encompassed the absence of atherosclerotic coronary disease, uncontrolled diabetes (defined as HbA1c>10%), a previous atherosclerotic cardiovascular event before age 75 (myocardial infarction, ischemic stroke, peripheral arterial disease, angina pectoris), a hematological malignancy (known or revealed by blood count at admission), known chronic inflammatory disease (rheumatoid arthritis, arteritis, gastro-intestinal illness), infection or fever >38.5 °C within 15 d prior to hospitalization, surgical operation within 30 d prior to hospitalization, the presence of an active neoplasia, and long-term use of anti-inflammatory therapies. Patients who were under legal protective measures, deprived of liberty by court order, unable to give their written informed consent, or having participated in an interventional study on a drug within 30 d before enrolment were also ineligible for enrollment in this study.

## Clinical and biological data reported in MI(+) and MI(-) subjects

Data related to patient's characteristics included: age, gender, and body mass index. For traditional alterable cardiovascular risk factors, smoking was defined as having an active smoking habit or having stopped within the last 3 y. Hypertension was defined as a systolic blood pressure of at least 140 mmHg and/or a diastolic blood pressure of at least 90 mmHg from the means of daily measures during the hospital stay, or by the use of anti-hypertensive drugs. Dyslipidemia was defined as a c-LDL >4.1 mmol/L (>1.6 g/L) and/or c-HDL <1.03 mmol/L (<0.4 g/L) in men and <1.29 mmol/L (<0.5 g/L) in women, or by the use of lipid-lowering drugs (*Arnett et al., 2019*). A threshold of HbA1c>6.5% was used to define diabetes, as well as the use of anti-diabetic agents (*American Diabetes Association, 2019*).

In MI(+) subjects, the following data were collected regarding the atherothrombotic event: the existence of a ST-elevation; concerned territory; the number of main vessels with a stenosis >50%, according to morphological assessment by the cardiologist performing the coronary angiography; acute phase treatment (medical treatment, fibrinolysis, angioplasty, coronary artery bypass graft

surgery); and presence and type of potential complications, including: heart failure (defined according to the 2016 ESC guidelines *Ponikowski et al., 2016*), ventricular arrhythmia (defined as sustained ventricular tachycardia or ventricular fibrillation), high-degree atrioventricular block, pericarditis, mechanical complication comprising mitral regurgitation, ventricular free-wall rupture, and ventricular septal defect, as well as intra-cardiac thrombosis diagnosed by echocardiography. A routine cardiovascular evaluation was performed at inclusion (between 2 and 7 mo after MI occurrence). Subjects were asked about the presence of dyspnea or angina, and evaluated with NYHA and CCS scales. Information was obtained from medical records about a potential recurrence of a cardiovascular event since the index event (new MI, coronary revascularization, stroke, hospitalization for acute heart failure). Traditional cardiovascular risk factors were noted. Routine biological analyses comprising blood count, high-sensitive CRP (hsCRP), lipid profile, and HbA1c were performed in the laboratory of the University Hospital of Bordeaux on fresh samples. A blood sample (EDTA) was drawn and sent within 4 hr to the Biological Cancer Resources Center of the University Hospital of Bordeaux.

For MI(-) subjects, data available at inclusion included hsCRP level, creatinine, lipid profile, and traditional CVRF. No blood count was available but none of the subjects developed cancer (including hematological malignancy) during follow-up suggesting that detectable somatic mutations were indicative of CHIP and not hematological malignancy.

## Measurement of atherosclerosis burden

For MI(+) subjects, a transthoracic echocardiography was performed at inclusion by trained cardiologists and ejection fraction was calculated. Supra-aortic trunks ultrasonography with 3D measurement of carotid atheroma volume was performed by trained physicians of the hospital University of Bordeaux, using a Philips iU22 probe equipped with a linear-3D volume convertor VL13-5 (Philips). Images were analyzed using the Vascular Plaque Quantification software on the QLAB 10.2 system (Philips). Carotid stenosis quantification was done with NASCET criteria. Functional ischemic testing was performed at the clinicians' discretion. For MI(-) subjects, atherosclerosis burden was assessed at the inclusion by ultrasound echography recording detection of atherosclerotic plaques, atherosclerotic plaque numbers, and intima-media thickness.

## Follow-up on subjects

For MI(+) subjects, the follow-up was conducted with a standardized questionnaire previously validated in clinical trials (*Lafitte et al., 2013*). Recurrence of MI as well as any significant cardiovascular event (cardiovascular death, acute coronary syndrome, stroke or transient ischemic attack, congestive heart failure, secondary coronary revascularization, or peripheral vascular surgery) occurring between the initial event and the year after inclusion in the study were recorded. All medical records of participants who died, or who reported on the questionnaire that they had experienced cardiovascular symptoms between baseline and follow-up evaluations, were reviewed by one of the investigators, and the patient practitioners were contacted. For MI(-) subjects, a follow-up was performed for up to 12 y. All cardiovascular events were recorded (including MI) with adjudication upon medical records by an expert committee.

## NGS sequencing and analysis

For both MI(+) and MI(-) subjects, DNA was extracted from total leukocytes obtained at inclusion. The search for somatic mutations was carried out by the Laboratory of Hematology of the University Hospital of Bordeaux using the Next Generation Sequencing (NGS) panel designed for the diagnosis and follow-up of myeloid hematological malignancies. The genes tested are detailed in *Supplementary file 1a*.

### Library preparation and sequencing

A custom RNA-baits panel was designed to cover 56 genes involved in myeloid malignancies. The list of the genomic regions targeted is available in the *Supplementary file 1a*. Libraries were prepared from 200 ng of DNA for each patient using SureSelect XT Low Input kit (Agilent) or using Magnis NGS Prep System (Agilent). Libraries were pooled for multiplex sequencing on a NextSeq550Dx (Illumina) with Mid Output Kit v2.

## Bioinformatic pipeline

We developed a bioinformatic pipeline to analyze sequencing data in order to control each step of analysis. FASTQ files were generated by bcl2fastq. They were then aligned against reference genome hg19 (2013) with bwa-mem, producing BAM files. After this step, duplicate reads were tagged but not removed using Agilent locatit software. Finally, coverage analysis was performed with bbctools, mosdepth, and samtools, and resulting metrics were gathered with MultiQC to assess sequencing data quality, including depth of coverage for every gene in the panel. For this study, we used three different tools for variant calling in order to detect with good accuracy mutations in samples. We have chosen GATK Mutect2, VarScan, and VarDict to detect somatic mutations. Annovar software (version 2020-06-08) and ensembl VEP (v.103) were used to annotate all called variants. The following databases were used:

> COSMIC 92 (Catalogue Of Somatic Mutations In Cancer)
> gnomAD 2.1.1 (The Genome Aggregation Database), ExaC (The Exome Aggregation Consortium), 1000 genome, ESP (Exome Sequencing Project) to assess variant frequency in world population
> SIFT, PolyPhen2, PROVEAN (in-silico pathology prediction tools)
> dbSNP (The Single Nucleotide Polymorphism database)
> ClinVar (2021-01-03) (information on the relationships between variants and human health): this descriptor provides information on the clinical effect of the variant
> InterVar (software tool for automatic clinical interpretation of genetic variants by the ACMG/AMP 2015 guideline).

We observed a median depth of sequencing of 2111 X [1578;2574] for all regions sequenced. For the 'prototypical' CHIP genes defined by *Vlasschaert et al., 2023a*, the sequencing depth was 2694 X [1875;3785] for patients from the CHAth study and 3455 X [2266;4885] for patients from the 3 C study. More specifically, for *DNMT3A* and *TET2* genes, the median depths of sequencing were 2531 X [1818;3313] and 3710 X [2444;4901] for patients from the CHAth and 3 C studies, respectively.

## Review and classification of mutations

Intronic and synonymous mutations were removed as well as variants with VAF <1% and variants with a minor allele frequency (MAF) ≥0.1% listed in databases (dbSNP, gnomAD, ExaC, 1000 genome, ESP). Variants known as recurring artifacts (manually curated and stocked in a local database) were also removed. Finally, retained variants were reviewed independently by two molecular biologists for (i) visual inspection of reads in the BAM file to determine if they were real mutations or artifacts and for (ii) classification of the pathogenicity of mutations. To be interpretable, the generated data had to meet the following criteria:

> Minimum read depth of 200 X.
> Sufficient coverage of the different regions in both reading directions.
> Total number of reads corresponding to the variant must be greater than 10.

In order to have more confidence in variants with low VAF, we distinguished them from artifacts by estimating the background noise *via* the median absolute deviation and the corresponding confidence intervals. To exclude variants that could represent artifacts, we only retained those that presented a VAF at least twofold higher than the upper limit of the confidence interval.

The classification of mutations was based on the consensus recommendation of the Association for Molecular Pathology and the American Society of Clinical Oncology (*Li et al., 2017*). Variants were classified as pathogenic, likely pathogenic, or of unknown significance according to the criteria shown in *Supplementary file 1b*, *Luque Paz et al., 2021*.

According to the definition of CHIP, only mutations with a VAF ≥2% were retained. Although we did not use error-corrected sequencing, the high depth (2111 X in median) coupled with bioinformatic tools and manual curing allowed us to reliably detect variants with a VAF ≥1%.

Although our NGS panel and criteria for the classification of mutations are classic for the study of hematological malignancies, it should be noticed that they do not strictly correspond to the criteria defined by *Vlasschaert et al., 2023b* who recently defined a list of genes/mutations considered to be prototypical CHIP-genes more tightly associated with cardiovascular disease.

## Detection of mLOY by digital droplets PCR

The search of mLOY was performed thanks to an in-house droplet digital PCR technique using the following primers and probes:

> Primer-amel-Fwd: 5'- CCCCTGGGCACTGTAAAGAAT
> Primer-amel-Rev: 5'- CCAAGCATCAGAGCTTAAACTG
> Probe-amelX: 5'- HEX-CCAAATAAAGTGGTTTCTCAAGT-BHQ
> Probe-amelY: 5'- FAM-CTTGAGAAACATCTGGGATAAAG-BHQ.

Briefly, 75 ng of DNA was mixed with ddPCR supermix for Probes (no dUTP, Biorad), primers (0.9 µM each), and probes (0.25 µM each). The emulsion was prepared with the QX-100 (Biorad). The amplification program was as follows: 10- min denaturation at 95 °C, followed by 40 cycles of 30 s at 94 °C, 1 min at 55 °C, and inactivation of 10 min at 98 °C. The number of droplets positive for amelX and amelY was determined on the QX-200 droplet reader (Biorad) using the Quantasoft software version 1.5 (Biorad). At least 10,000 droplets were analyzed in each well. We determined the background noise of our technique by analyzing the DNA of control subjects (men under 40 y old with a normal karyotype, as assessed by conventional cytogenetic studies). We observed that only a signal corresponding to 9% of cells with mLOY could be considered different from background noise. By analyzing a dilution series of control DNA, we demonstrated the reliability of our ddPCR assay in estimating the proportion of cells with mLOY and its ability to detect as low as 10% of cells with mLOY. Considering the background noise, we established our threshold for confirming the presence of mLOY at 9% of cells with mLOY. We considered the level of mLOY as 'low' when the proportion of cells with a mLOY was between 9 and 50%, and a 'high' when the proportion of cells with mLOY was above 50%.

**Table 1.** Subjects' characteristics.

| | All subjects n=446 | MI(+) subjects n=149 | MI(-) subjects n=297 | p-value |
|---|---|---|---|---|
| Male, n (%) | 257 (57.6) | 98 (66) | 159 (54) | 0.015 |
| Median age, years (Q1;Q3) | 76.4 (71.9;80.9) | 82.0 (78.0;86.0) | 73.6 (70.6;77.8) | $p<10^{-4}$ |
| Cardiovascular risk factors | | | | |
| BMI, kg/m² (Q1;Q3) | 25.5 (23.6;28.3) | 25.5 (23.6;28.5) | 25.5 (23.6;28.1) | 0.14 |
| Diabetes, n (%) | 94 (21.2) | 47 (32%) | 47 (16%) | $p<10^{-4}$ |
| Hypertension, n (%) | 362 (82.8) | 107 (76.4) | 255 (85.9) | 0.020 |
| Total cholesterol, g/L (Q1;Q3) | 2.08 (1.72;2.38) | 1.45 (1.25;1.72) | 2.23 (2.00;2.47) | $p<10^{-4}$ |
| LDL-c, g/L (Q1;Q3) | 1.25 (0.94;1.52) | 0.77 (0.61;1.03) | 1.39 (1.19;1.60) | $p<10^{-4}$ |
| HDL-c, g/L (Q1;Q3) | 0.56 (0.46;0.66) | 0.47 (0.39;0.57) | 0.59 (0.50;0.68) | $p<10^{-4}$ |
| Smoking, n (%) | 32 (7.2) | 6 (4.4) | 26 (8.7) | 0.117 |
| Prevalence of CHIP and mLOY | | | | |
| CHIP prevalence, n (%) | 201 (45.1) | 79 (53%) | 122 (41%) | 0.923 |
| Prevalence of CHIP with VAF ≥5% | 88 (19.7) | 30 (20.1) | 58 (19.5) | 0.069 |
| Subjects tested for mLOY, n | 220 | 97 | 123 | - |
| mLOY prevalence, n (%) | 83 (37.7) | 44 (45.4) | 39 (31.7) | 0.783 |
| CHIP(+) / mLOY(+) prevalence, n (%) | 39 (17.7) | 22 (22.7) | 17 (13.8) | 0.797 |

Data are expressed as numbers and frequency or median, first and third quartiles. For quantitative values, comparisons were made by linear regression of log values adjusted for age and sex. For qualitative parameters, comparisons were made by the fisher test. For each variable, results are expressed among patients with available value.

### Statistical analyses

Univariate association analyses were conducted using Fisher exact or Chi-square test statistics for categorical variables. Analysis of variance was used for quantitative variables. Multivariate association analyses were performed using logistic or linear regression models as appropriate. Analyses were adjusted for age and sex (CHIP) or for age only (mLOY). We used raw values for all quantitative variables as they presented a normal distribution, except for CRP levels for which we analyzed log(CRP). Log-rank tests and Cox statistical models were employed to assess the association of clinical/biological variables with the incidence of future cardiovascular events.

All analyses were conducted using either the RStudio software (Posit team (2023). RStudio: Integrated Development Environment for R. Posit Software, PBC, Boston, MA. URL http://www.posit.co/.) or the PRISM software (GraphPad Prism version 9.5.1).

## Results

### Patient's characteristics

In this study, we aimed to decipher whether mLOY could alter the effect of CHIP in inducing a chronic inflammation that would favor the development of atherosclerosis and the incidence of atherothrombotic events. To answer this question, we analyzed 446 subjects from two prospective studies, 149 who presented a MI at inclusion and 297 who did not present any CVE before inclusion. The general

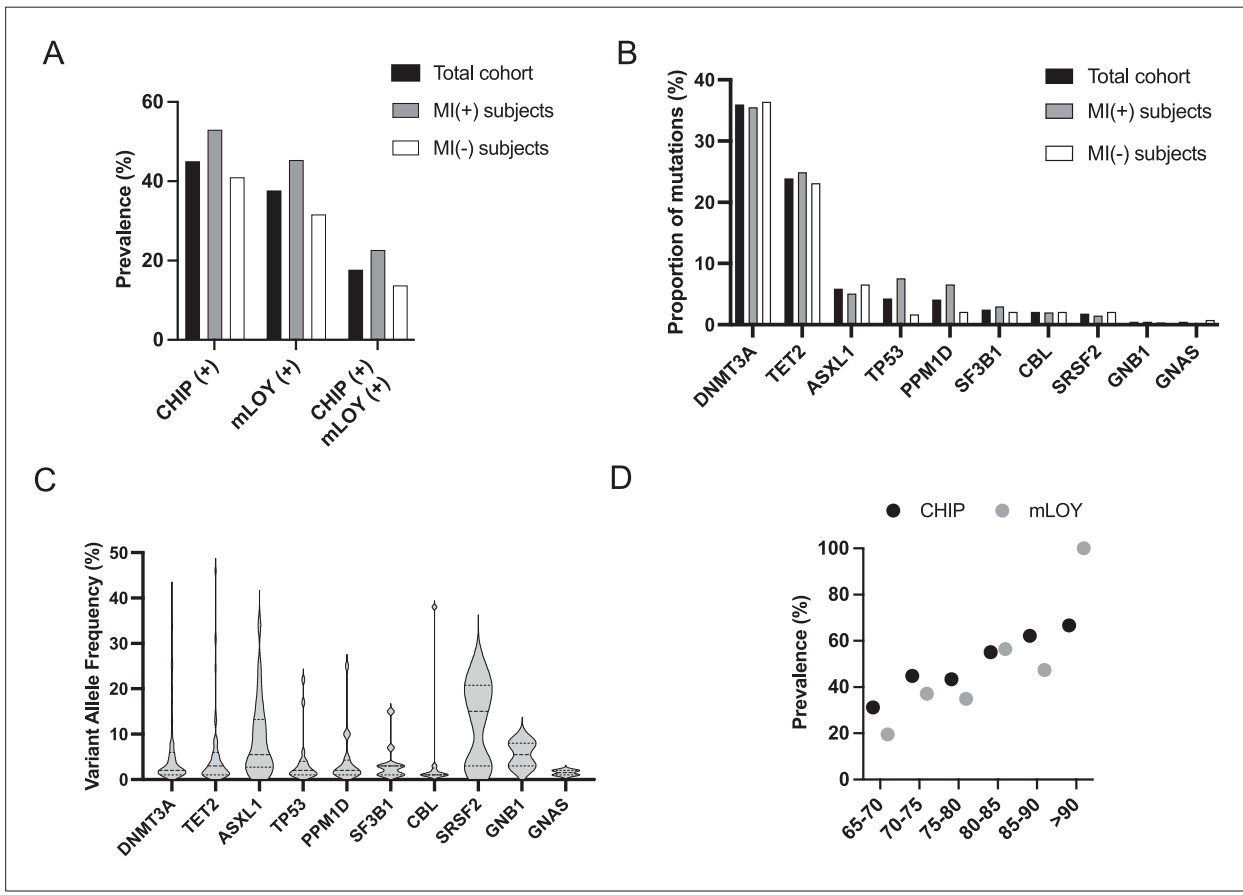

**Figure 1.** Clonal hematopoiesis of indeterminate potential (CHIP), mosaic loss of Y chromosome (mLOY) and their combination are as frequent in myocardial infarction (MI)(+) and MI(-) subjects. (**A**) Prevalence of CHIP, mLOY, and their combination in the total cohort of 449 subjects, in MI(+) as well as in MI(-) subjects. (**B**) Mutational spectrum of CHIP expressed as the proportion of mutations detected in the indicated genes. (**C**) Variant allele frequency (VAF) measured for the different mutations detected in the 449 subjects detected in the indicated genes. (**D**) Prevalence of CHIP and mLOY depending on age in the total cohort (for CHIP) and in male subjects (for mLOY).

The online version of this article includes the following figure supplement(s) for figure 1:

**Figure supplement 1.** Variant allele frequency (VAF) of somatic mutations detected in myocardial infarction (MI)(+) and MI(-) subjects.

characteristics of the 446 combined subjects are detailed in *Table 1*. Briefly, the median age was 76.4 y and 257 (57.6%) were males. Forty percent of them presented more than 2 CVRF. MI(+) subjects were older (due to the inclusion criteria), more frequently men, and presented a lower cardiovascular risk than MI(-) subjects (due to the initiation of treatment between the initial event and the inclusion).

## CHIP and mLOY are detected as frequently in MI(+) and MI(-) subjects

Among the 446 subjects, at least one mutation with a VAF ≥2% was detected in 201 persons (45.1%), defining CHIP(+) subjects (*Table 1*, *Figure 1A*). As previously described, *DNMT3A* and *TET2* were the two most frequently mutated genes. The other mutated genes were those previously described in CHIP (*Figure 1B*, *Jaiswal et al., 2017*; *Jaiswal et al., 2014*; *Zink et al., 2017*) and the median VAF was 2% (*Figure 1C*). The mutational profile of CHIP(+) subjects is available in the *Supplementary file 1c and d* while the characteristics of CHIP(+) compared with CHIP(-) subjects are detailed in the *Supplementary file 1e and f*. In this study, we considered subjects without any detectable mutation or with only mutations with a VAF below 2% as non-CHIP carriers (CHIP(-) subjects).

A mLOY was present in 83 (37.7%) male subjects (mLOY(+) subjects, *Table 1*, *Figure 1A*). The clinico-biological characteristics of mLOY(+) subjects compared with mLOY(-) subjects are detailed in the *Supplementary file 1e and f*. The median proportion of cells with mLOY was 18% [12%;32%]. There was no significant association between CHIP and mLOY since 39 CHIP(+) subjects (39.8%) also carried a mLOY compared with 44 CHIP(-) subjects (36.1%, p=0.579). Finally, as previously demonstrated, CHIP(+) subjects were significantly older than CHIP(-) subjects (*Supplementary file 1e and f*). We also observed a significant association between age and CHIP prevalence (p<0.001). Similarly, the prevalence of mLOY increased with age (p<0.001, *Figure 1D*).

We observed a similar frequency of CHIP and mLOY in MI(+) and MI(-) subjects (*Figure 1A*, *Table 1*). Besides, the association of CHIP with mLOY was as frequent in MI(+) and MI(-) subjects (22.7% and 13.8%, p=0.797). Of note, MI(+) and MI(-) subjects presented similar proportions of mutated genes (*Figure 1B*) and VAF (*Figure 1—figure supplement 1*). Similar results were also observed when considering CHIP associated with important clones (i.e. VAF ≥5%, *Table 1*), when analyzing only *DNMT3A* and *TET2* mutations (data not shown), or when making further adjustments on CVRF.

Altogether, these results suggest that CHIP and mLOY are very frequent but not associated with the existence of a history of MI, even when mLOY is associated with CHIP.

## Neither CHIP, mLOY, nor their combination highly increase basal inflammation or atherosclerotic burden

It was demonstrated that *TET2* mutations were associated with the induction of a pro-inflammatory phenotype of macrophages (*Fuster et al., 2017*). On the contrary, mLOY was shown to decrease the inflammatory phenotype of macrophages (*Sano et al., 2022*). Therefore, we searched to determine whether mLOY could counter the inflammatory state that would be associated with CHIP. In the total cohort, CHIP(+) and CHIP(-) subjects presented similar levels of hs-CRP, as did mLOY(+) and mLOY(-)

**Table 2.** .CHIP and mLOY are not associated with increased hsCRP level.

| | hsCRP level All subjects | p value | hsCRP level MI(+) subjects | p value | hsCRP level MI(-) subjects | p value |
|---|---|---|---|---|---|---|
| CHIP(-) | 1.64 (1.00;3.69) | 0.652 | 1.40 (1.00;4.00) | 0.600 | 1.71 (0.97;3.22) | 0.141 |
| CHIP(+) | 2.00 (1.00;3.90) | | 2.20 (1.10;5.00) | | 1.63 (0.91;2.54) | |
| mLOY(-) | 1.45 (0.99;2.75) | 0.156 | 1.8 (1.0;4.8) | 0.149 | 1.41 (0.74;2.16) | 0.358 |
| mLOY(+) | 1.73 (1.01;4.00) | | 2.4 (1.03;4.5) | | 1.2 (0.99;2.99) | |
| CHIP (-) mLOY (-) | 1.11 (0.76;2.19) | 0.410 | 1.00 (0.77;2.72) | 0.430 | 1.35 (0.79;2.14) | 0.570 |
| CHIP (+) mLOY (-) | 1.87 (1.00;3.03) | | 2.30 (1.35;7.35) | | 1.43 (0.72;2.37) | |
| CHIP (-) mLOY (+) | 2.20 (1.02;4.00) | | 2.50 (1.30;4.00) | | 1.37 (0.95;3.75) | |
| CHIP (+) mLOY (+) | 1.23 (1.01;3.80) | | 2.00 (1.02;4.75) | | 1.17 (1.03;1.73) | |

hsCRP Data are expressed as median, first and third quartiles. Comparisons were performed by linear regression of log values adjusted for age and sex. hsCRP levels are expressed in mg/L For each variable, results are expressed among patients with available value.

**Table 3.** CHIP and mLOY are not associated with an increased atherosclerotic burden.

**Atherosclerosis burden evaluation in MI(+) subjects**

| | All patients (n=149) | CHIP (-) (n=70) | CHIP (+) (n=79) | p-value | mLOY (-) (n=53) | mLOY (+) (n=44) | p-value |
|---|---|---|---|---|---|---|---|
| Multitroncular lesions, n (%) | 68 (45.6) | 29 (41.4) | 39 (49.4) | 0.484 | 25 (47.2) | 20 (45.4) | 0.717 |
| Carotid stenosis ≥50%, n (%) | 7 (4.7) | 2 (2.8) | 5 (6.3) | 0.317 | 2 (3.8) | 3 (6.8) | 0.451 |
| Global atheroma volume (mm³), median (Q1;Q3) | 499.5 (408.0;604.5) | 455.0 (374.0;555.0) | 520.0 (411.5;611.5) | 0.333 | 601.0 (412.0;718.0) | 492.0 (344.5;600.5) | 0.707 |

**Atherosclerosis burden evaluation in MI(-) subjects**

| | All patients (n=297) | CHIP (-) (n=175) | CHIP (+) (n=122) | p-value | mLOY (-) (n=84) | mLOY (+) (n=39) | p-value |
|---|---|---|---|---|---|---|---|
| Patients with atherosclerotic plaque, n (%) | 135 (45.4) | 81 (46.3) | 54 (44.3) | 0.997 | 34 (40.5) | 19 (48.7) | 0.537 |
| Number of plaque, median (Q1;Q3) | 1 (1;2) | 2 (1;2) | 1 (1;2) | 0.258 | 2 (1;2) | 2 (1;2) | 0.863 |
| Intima Media Thickness (mm), median (Q1;Q3) | 0.68 (0.60;0.76) | 0.67 (0.60;0.76) | 0.68 (0.59;0.74) | 0.897 | 0.67 (0.62;0.76) | 0.72 (0.57;0.83) | 0.706 |

Data are expressed as numbers and frequency or median, first and third quartiles. For quantitative values, comparisons were made by linear regression of log values adjusted for age and sex. For qualitative parameters, comparisons were made by the fisher test and logistic regression. For each variable, results are expressed among patients with available value.

subjects (*Table 2*). Similarly, hs-CRP levels were not different in CHIP(-)/mLOY(-), CHIP(+)/mLOY(-), CHIP(-)/mLOY(+) and CHIP(+)/mLOY(+) subjects.

The impact of CHIP and mLOY on hsCRP levels, either alone or in combination, was comparable in MI(+) or MI(-) subjects (*Table 2*). This was also true when restricting the analysis to *DNMT3A* and *TET2* mutated CHIP(+) subjects (*Supplementary file 1g*), or when adjusting further on CVRF. Finally, subjects with important CHIP or mLOY clones did not present higher levels of hs-CRP (*Supplementary file 1h*).

Very few data are available about the atheroma burden associated with CHIP and/or mLOY in humans. Moreover, the modulation of the atherogenic effect of CHIP by mLOY remains unexplored. We thus asked whether CHIP alone or in association with mLOY was associated with an increased atherosclerotic burden. In MI(+) subjects we observed a similar proportion of multitroncular coronary lesions and carotid stenosis >50% in CHIP(+) and CHIP(-) subjects (*Table 3*). The global atheroma volume was explored by 3D ultrasound in 34 MI(+) patients, without difference between CHIP(+) and CHIP(-) subjects. Similar results were obtained when analyzing only CHIP(+) subjects carrying *TET2* and/or *DNMT3A* mutations (*Supplementary file 1g*), when comparing mLOY(+) and mLOY(-) subjects (*Table 3*), or when analyzing the association of CHIP with mLOY (*Supplementary file 1i*). Concordantly, in MI(-) subjects, all available atherosclerosis markers were similar between CHIP(+) and CHIP(-), as well as between mLOY(+) and mLOY(-) subjects (*Table 3*). This was also true when analyzing the effect of the different combinations of CHIP and mLOY (*Supplementary file 1i*). Once again, all these results were confirmed in subjects with a VAF ≥5% for CHIP or a mLOY ≥50% (*Supplementary file 1h*), or when adjusting further on CVRF.

Altogether, our results suggest that in both the context of a recent MI and in healthy individuals, CHIP is not associated with a systemic inflammation or an increased atherosclerotic burden. Additionally, mLOY does not modulate inflammatory parameters or atherosclerosis, even in the presence of CHIP.

## Neither CHIP, mLOY, nor their combination highly impact the incidence of MI

To decipher whether mLOY could impact the atherothrombotic risk associated with CHIP, we analyzed the incidence of MI during the follow-up of MI(-) subjects. Seventy-nine subjects developed a MI after inclusion in the study with a median delay of 5.00 years. Subjects with MI during follow-up did not differ significantly from those without MI in terms of demography or cardiovascular risk (*Supplementary file 1j*). Contrary to other studies, we did not observe an association between CHIP and an increased incidence of MI (HR 1.033 [0.657;1.625] after adjustment on age, sex, and CVRF). In comparison, hypercholesterolemia and smoking tended to be associated with a stronger risk of incident MI (HR = 1.474 [0.758;2.866] and 1.866 [0.943;3.690], respectively). Similarly, neither mLOY nor the association

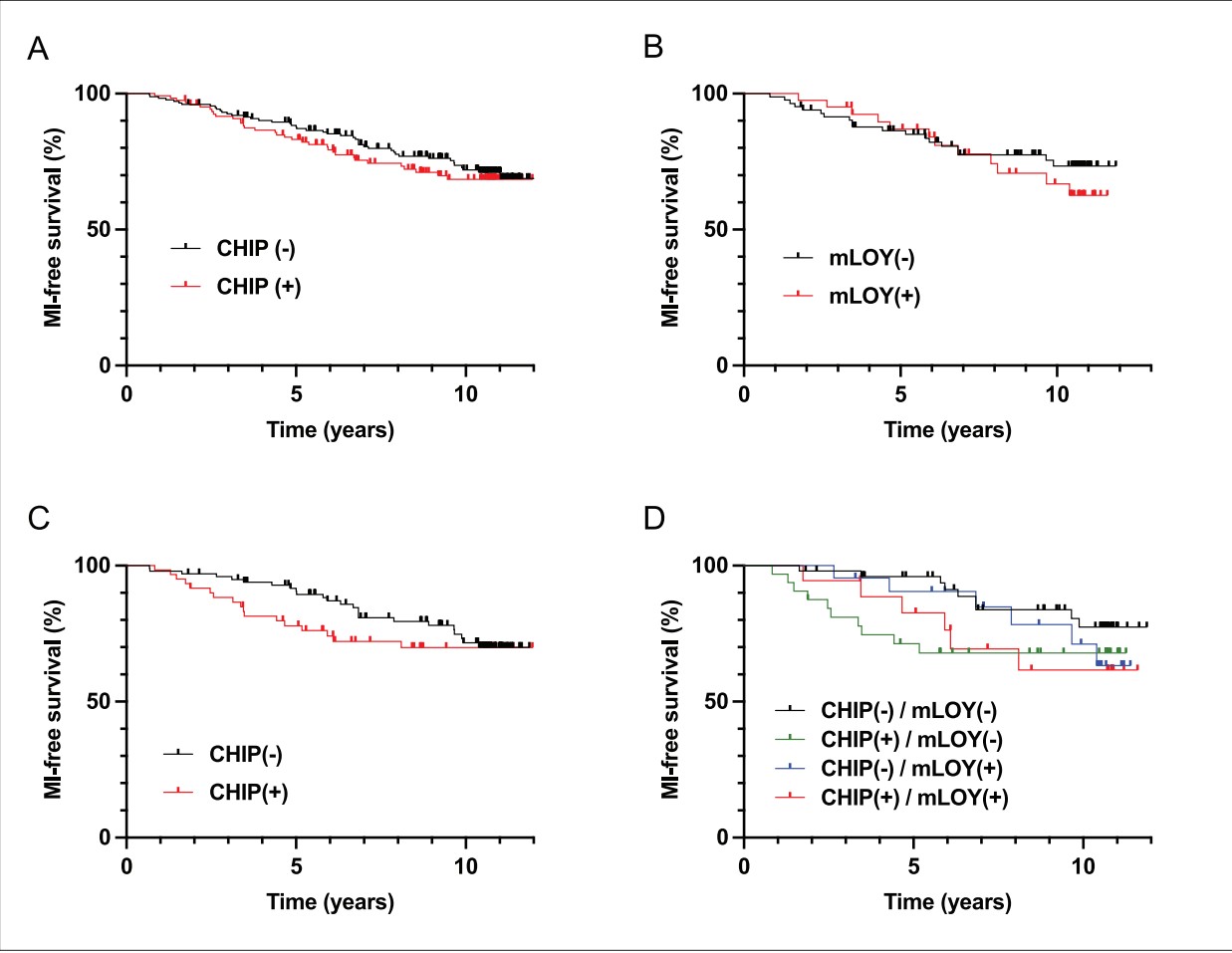

**Figure 2.** Clonal hematopoiesis of indeterminate potential (CHIP) and mosaic loss of Y chromosome (mLOY) do not increase significantly the risk of incident myocardial infarction (MI), but could accelerate it in male subjects. Incidence of MI during follow-up according to the presence of CHIP (**A**) or mLOY (**B**) in MI(-) subjects. Incidence of MI during follow-up according to the presence of CHIP (**C**) or the combination of CHIP and mLOY (**D**) in male MI(-) subjects. Survival was compared between the different groups with log-rank tests.

The online version of this article includes the following figure supplement(s) for figure 2:

**Figure supplement 1.** Clonal hematopoiesis of indeterminate potential (CHIP) do not increase significantly the risk of incident myocardial infarction (MI).

between CHIP and mLOY were associated with an increased incidence of MI (*Figure 2A–B*, *Supplementary file 1k*). Concordantly, we did not observe any difference in the prevalence of CHIP, mLOY, or their association between MI(-) subjects who suffered from a MI during follow-up and those who did not (*Supplementary file 1j*). This was also the case when restricting the analysis to CHIP with a VAF ≥5%, to subjects with a proportion of cells with mLOY ≥50%, to CHIP associated with *DNMT3A* or *TET2* mutations (*Figure 2—figure supplement 1*, *Supplementary file 1j*), or when making further adjustment on CVRF. These results suggest that the atherothrombotic risk associated with CHIP is moderate, and is not modulated by its association with mLOY. Moreover, neither CHIP, mLOY nor their combination were significantly associated with atherothrombotic recurrence (*Supplementary file 1k*).

## CHIP in the absence of mLOY may accelerate the occurrence of MI

In order to search for a combined effect of CHIP with mLOY on the risk of incidence of MI, we finally focused our analysis on MI(-) male subjects. In this population, we observed that MI occurred earlier in CHIP(+) subjects, with a significantly increased 5 y incidence of MI (log-rank test, p=0.014, *Figure 2C*). Such an effect was not observed in MI(-) female subjects (log-rank test, p=0.9402, *Figure 2—figure supplement 1*). Interestingly, this effect was more pronounced in CHIP(+)/mLOY(-) subjects who

presented a significantly higher 5 y incidence of MI (log-rank test, p=0.010, *Figure 2D*) and a significantly lower median time to MI compared with other subjects (Kruskall-Wallis test, p=0.007). CHIP(+)/mLOY(-) subjects also presented a higher 5 y incidence of MI using Cox models after adjustment on age and CVRF (HR 7.81, p=0.0388). Altogether, our results suggest that CHIP does not increase the risk of MI, but may accelerate its incidence, particularly in the absence of mLOY.

## Discussion

CHIP has previously been implicated in decreased overall survival, primarily due to its association with an increased incidence of cardiovascular diseases such as coronary artery disease (CAD) and stroke (*Jaiswal et al., 2017*; *Jaiswal et al., 2014*). Experimental models have suggested that this association may arise from the pro-inflammatory phenotype of mutated monocytes/macrophages, contributing to the development of atherosclerotic plaques (*Fuster et al., 2017*; *Jaiswal et al., 2017*). However, recent findings in the literature have presented a complex and sometimes contradictory picture. Studies have reported varying results regarding the relationship between CHIP, inflammation markers, and atherothrombotic events, challenging the initial notion of a high atherothrombotic risk associated with CHIP (*Bick et al., 2020*; *Busque et al., 2020*; *Kar et al., 2022*; *Kessler et al., 2022*; *Vlasschaert et al., 2023a*). Moreover, there has been a scarcity of evidence linking CHIP to an increased burden of atherosclerosis in human subjects (*Heimlich et al., 2024*; *Jaiswal et al., 2017*; *Wang et al., 2022*; *Zekavat et al., 2023*). Additionally, limited data are available on the potential impact of chromosomal abnormalities, particularly mLOY, on atherothrombosis in individuals with CHIP. In this study, we sought to investigate whether mLOY could modulate the effects of CHIP concerning systemic inflammation, atherosclerotic burden, and the risk of atherothrombotic events. To achieve this, we employed sensitive techniques, including targeted high-throughput sequencing and digital PCR, to analyze samples from two cohorts of meticulously phenotyped subjects.

In contrast to many previous studies, we conducted an analysis involving two distinct cohorts. The 'cases' were individuals recruited from the CHAth study within 2–7 mo following their first MI after the age of 75. We also established a 'control cohort' comprising 297 subjects from the 3 C cohort, none of whom had experienced CVE before inclusion. This allowed us to assess the effects of CHIP and mLOY on inflammation and atherosclerosis independently of pre-existing cardiovascular disease. Our analysis revealed a remarkably high frequency of CHIP, with an estimated prevalence of 45% among our 446 subjects, with a median age of 76.4 y. This prevalence exceeded initial reports of 15–20% determined by whole exome sequencing (WES) in individuals aged 70–80 (*Genovese et al., 2014*; *Jaiswal et al., 2014*), likely attributable to the enhanced sensitivity of our sequencing technique. Importantly, our estimated prevalence aligns with studies employing similarly sensitive high-throughput sequencing techniques .(*Guermouche et al., 2020*; *Mas-Peiro et al., 2020*; *Pascual-Figal et al., 2021*; *van Zeventer et al., 2021*). Furthermore, our approach allowed us to reliably detect mutations with a VAF as low as 1%. We chose to define subjects with detectable mutations at a VAF of 1% as CHIP(-), aligning with the WHO definition of CHIP (*Khoury et al., 2022*). Notably, all our analyses also yielded identical results when comparing CHIP(+) patients to subjects without any detectable mutations or when comparing all patients with somatic mutations with a VAF ≥1% to those without.

A recent study by Mas-Peiro et al. demonstrated an association between mLOY and increased post-transcatheter aortic valve replacement (TAVR) mortality (*Mas-Peiro et al., 2023*). In their investigation, a mLOY ratio threshold of 17% was deemed the most relevant for discriminating patients' mortality risk. In our study, we employed a mLOY threshold of 9% to identify mLOY, a value determined to allow a reliable mLOY detection and to distinguish it from the background noise signal. Using this threshold, we frequently detected mLOY in male subjects within our cohort, with an estimated prevalence of 37.7% among 220 subjects. This prevalence surpasses that reported by *Forsberg et al., 2014* but aligns with recent findings from studies employing sensitive techniques (*Zink et al., 2017*).

Surprisingly, our cohort did not reveal any significant association between the presence of CHIP and the detection of mLOY. This contrasts with the results of two recent studies (*Ljungström et al., 2022*; *Zink et al., 2017*) possibly explained by the heightened sensitivity of our methodology in reliably detecting both somatic mutations and mLOY, which may exist in very small proportions of blood cells.

Mouse models of CHIP have recently suggested that somatic mutations are linked to a pro-inflammatory phenotype of mutated monocytes/macrophages, an observation supported by human samples using single-cell RNA sequencing (*Abplanalp et al., 2021*). However, contradictory results have emerged when examining plasma markers of inflammation. Studies, including ours, have not consistently identified a significant increase in plasma hs-CRP or proinflammatory cytokines in CHIP carriers (*Cook et al., 2019*; *Pascual-Figal et al., 2021*), whereas others have reported such associations (*Bick et al., 2020*; *Busque et al., 2020*). Importantly, our study did not show any significant association between the detection of CHIP(+/-)mLOY and plasma levels of IL1ß and IL6 in MI(+) subjects, suggesting that CHIP's association with increased systemic inflammation may depend on specific stimulating factors. Notably, recent studies have reported elevated levels of plasma inflammatory markers in CHIP carriers during atherothrombotic events, such as MI or stroke (*Arends et al., 2023*; *Böhme et al., 2022*; *Wang et al., 2022*), indicating that CHIP may amplify systemic inflammation under specific conditions, but not necessarily in a basal state, as in our study.

To date, only a limited number of studies have successfully linked the presence of CHIP to an increased atherosclerotic burden. While *Jaiswal et al., 2017* reported a higher calcic score in subjects with CHIP, and *Zekavat et al., 2023* suggested an increased atherosclerosis, these evaluations relied on self-reported atheroma and indirect parameters. In the context of coronary artery disease, only 2 studies addressed this question with conflicting results. *Heimlich et al., 2024* observed an increased prevalence of stenosis and obstructive stenosis in CHIP(+) subjects, particularly of the left main artery, while *Wang et al., 2022* did not notice any association between CHIP and the extent of coronary artery disease. Our study stands as one of the first to utilize direct markers of atherosclerosis, including global atheroma volume, in CHIP(+) subjects within the context of coronary artery disease. Strikingly, we did not detect a clear increase in atherosclerosis among CHIP or mLOY carriers, either individually or in combination. Conversely, increased atherosclerotic burden associated with CHIP has been observed in patients with stroke (*Mayerhofer et al., 2023*).

Our study bears some limitations, the first of them being a relatively modest sample size of 449 subjects, which did not allow us to establish a direct association between CHIP, either alone or in conjunction with mLOY, and coronary heart disease. These results are in contradiction with previous studies based on cohorts composed of a high number of subjects (*Jaiswal et al., 2014*; *Jaiswal et al., 2017*; *Vlasschaert et al., 2023a*). At the time of our study's initiation, the literature suggested a CHIP prevalence of 20% after 75 y and an increased MI risk associated with CHIP, with a hazard ratio of 1.7. Based on this data, a cohort of 112 cases would have been sufficient to demonstrate a more frequent presence of CHIP in MI(+) patients compared to MI(-) subjects with a power of 0.90 (more details are available in the Supplementary methods). Although our study was not designed to demonstrate an association between CHIP and incident MI, we were able to confirm that the increased risk of MI associated with the presence of CHIP, if any, is lower than 1.7, which is in accordance with more recent studies (*Vlasschaert et al., 2023a*; *Zekavat et al., 2023*; *Zhao et al., 2024*).

Different parameters could have also contributed to the discrepancy of our results with those of previous studies. First, the age of our subjects (≥75 y in the CHAth study,≥65 y in the 3 C study) is higher than the one of the other cohorts (*Jaiswal et al., 2017*; *Vlasschaert et al., 2023a*; *Zhao et al., 2024*). Then our strategy to search for somatic variants was also different. In particular, we did not use the criteria defined by *Vlasschaert et al., 2023b* to cure variants that were called. This had a limited effect since 86.8% of the variants detected in our cohort were concordant with the criteria of *Vlasschaert et al., 2023b*, impacting the conclusion on the existence of a CHIP in only 15 patients. We also searched for a specific effect of *TET2* mutations on inflammation, atherosclerosis, or risk of MI, as the literature suggests that they could be considered as 'positive controls.' However, we did not find any significant effect of *TET2* mutations. Finally, we were not able to reliably detect variants with a VAF <1% and could have missed the effect of low-VAF variants, as recently shown by *Zhao et al., 2024*.

However, our results align with previous studies that reported either no difference in CHIP prevalence between individuals with MI and those without (*Busque et al., 2020*), or no significant association between CHIP and incident de novo or recurrent atherothrombotic events (*Arends et al., 2023*; *David et al., 2022*; *Kar et al., 2022*). Recently, *Kessler et al., 2022* proposed that the increased risk of atherothrombotic events might be limited to CHIP with a VAF of 10% or higher. However, even when considering this criterion, the association was not validated in a cohort of 173,585 subjects.

Moreover, in our cohort the observed effect of CHIP on the risk of MI (HR=1.033) was substantially lower than the ones observed for other established cardiovascular risk factors such as hypercholesterolemia (HR=1.475) or smoking (HR=1.865). This underscores the formidable challenges of identifying associations with low effect sizes, necessitating cohorts comprising hundreds of thousands of subjects to achieve statistical significance. Collectively, these findings suggest that any atherothrombotic risk associated with CHIP is limited in scope and cannot be used in clinical practice for the management of patients with a history of cardiovascular disease or a risk of atherothrombosis.

However, we believe that our study has also some strengths. Notably, we employed highly sensitive techniques for the reliable detection of both CHIP and mLOY, surpassing the capabilities of large cohort studies relying on whole exome sequencing and SNP arrays. Furthermore, we meticulously assessed atherosclerotic burden using sensitive parameters and evaluated two cohorts—one comprising MI(+) subjects and the other MI(-) subjects—with precise cardiovascular phenotyping at inclusion and rigorous follow-up, including direct evaluation and adjudication of all CVE.

In summary, our findings provide a nuanced perspective on the relationship between CHIP, mLOY, and cardiovascular outcomes, highlighting the need for larger, more comprehensive studies to elucidate potential associations further. Our findings that CHIP might expedite the onset of MI, particularly in the absence of mLOY warrant further investigation in larger subject cohorts.

## Acknowledgements

The authors would like to thank the University Hospital of Bordeaux CRB-K for processing the samples and Coralie Foucault and the Agilent Society for their partnership in supplying NGS reactive. We also thank Marina Migeon, Joël Decombe, and Candice Falourd for their technical help, as well as Christel Duprat and Matthieu Meilhan for the operational organization of the project.

## Additional information

### Funding

| Funder | Grant reference number | Author |
|---|---|---|
| Fondation Coeur Recherche | | Thierry Couffinhal |
| Federation Francaise de Cardiologie | | Olivier Mansier |
| ERA-CVD | JTC 2019 | Chloe James |

The funders had no role in study design, data collection and interpretation, or the decision to submit the work for publication.

### Author contributions

Sami Fawaz, Data curation, Investigation, Writing – original draft; Severine Marti, Melody Dufossee, Data curation, Formal analysis, Investigation, Writing – original draft; Yann Pucheu, Astrid Gaufroy, Jean Broitman, Audrey Bidet, Investigation; Aicha Soumare, Christophe Tzourio, Stephanie Debette, Project administration; Gaëlle Munsch, Methodology; David-Alexandre Trégouët, Data curation, Methodology, Writing – original draft, Writing – review and editing; Chloe James, Conceptualization, Writing – review and editing; Olivier Mansier, Conceptualization, Data curation, Formal analysis, Funding acquisition, Investigation, Writing – original draft, Project administration, Writing – review and editing; Thierry Couffinhal, Conceptualization, Funding acquisition, Investigation, Writing – original draft, Project administration, Writing – review and editing

### Author ORCIDs

Gaëlle Munsch ⓘ https://orcid.org/0000-0003-1564-9825
Christophe Tzourio ⓘ https://orcid.org/0000-0002-6517-2984
David-Alexandre Trégouët ⓘ https://orcid.org/0000-0001-9084-7800
Olivier Mansier ⓘ https://orcid.org/0000-0002-7943-8800

### Ethics

Clinical trial registration NCT04581057.

All participants gave written informed consent before inclusion in the study. The study was approved by the institutional review board (IRDCB 2019-A02902-05), and registered (https://www.clinicaltrials.gov : NCT04581057).

Reviewer #2 (Public review): https://doi.org/10.7554/eLife.96150.3.sa1

Author response https://doi.org/10.7554/eLife.96150.3.sa2

---

## Additional files

### Supplementary files

• Supplementary file 1. Additionnal data about genetic, inflammatory and atoscletrotic parameters in the studied subjects. (a) Targeted sequencing panel used in the laboratory of Hematology of the University Hospital of Bordeaux for the diagnosis and follow-up of myeloid hematological malignancies and used in this study. (b) Characteristics of the identified mutations according to the degree of certainty of their pathogenicity (deleterious, possibly deleterious, or variant of undetermined significance). Exceptions: ASXL1/ASXL2/TET2: only variants resulting in a truncated protein were retained as deleterious variants (A). For TET2, missense mutations were retained only when they affected CD1 or CD2 domain ANKRD26: non-coding region 5'-UTR was retained. TP53: database IARC (https://p53.iarc.fr/) was used for the classification. * Only variants classified as A or B are retained as clonal hematopoiesis of indeterminate potential (CHIP) in the study. (c) Mutational profile of CHIP(+) MI(+) subjects. (d) Mutational profile of CHIP(+) MI(-) subjects. (e) Comparison of characteristics of MI(+) subjects depending on their CHIP and mosaic loss of Y chromosome (mLOY) status. (f) comparison of characteristics of MI(-) subjects depending on their CHIP and mLOY status. For statistical analysis, logistical regression (adjusted on age and sex) and/or Fisher's test were used to compare qualitative variables, and linear regression (adjusted on age and sex) and/or ANOVA were used to compare quantitative variables. Blood counts were not available for MI(-) subjects. (g) CHIP associated with DNMT3A or TET2 mutations do not present differential effect on inflammation or atherosclerotic burden. Data are expressed as numbers and frequency or median, first, and third quartiles. For quantitative values, comparisons were made by linear regression of log values adjusted for age and sex. For qualitative parameters, comparisons were made by the fisher test. (h) Important clones of CHIP and mLOY are not associated with increased inflammation, atherosclerosis, or CVE. Data are expressed as numbers and frequency or median, first, and third quartiles. For quantitative values, comparisons were made by linear regression of log values adjusted for age and sex. For qualitative parameters, comparisons were made by the Fisher test and logistic regression. For each variable, results are expressed among patients with available values. (i) mLOY do not impact the atherosclerotic burden in the presence of CHIP. Data are expressed as numbers and frequency or median, first and third quartiles. For quantitative values, comparisons were made by linear regression of log values adjusted for age and sex. For qualitative parameters, comparisons were made by the Fisher test. For each variable, results are expressed among patients with available values. (j) Characteristics of MI(-) patients at the time of inclusion depending on whether they presented a myocardial Infarction (MI) during follow-up or not. Data are expressed as numbers and frequency or median, first and third quartiles. For quantitative values, comparisons were made by linear regression of log values adjusted for age and sex. For qualitative parameters, comparisons were made by the Fisher test and logistic regression. For each variable, results are expressed among patients with available values. (k) CHIP may accelerate the incidence MI in the absence of mLOY in male subjects. Data are expressed as numbers and frequency or median, first, and third quartiles. For the 'time to MI' in MI(-) subjects, comparisons were performed by the Mann-Whitney test for CHIP and mLOY separately and the Kruskall-Wallis test for combinations of CHIP +/- mLOY. For qualitative parameters, comparisons were made by the Fisher test. For each variable, results are expressed among patients with available values.

• MDAR checklist

### Data availability

Data for the somatic variants are detailed in supplementary files 1c and 1d. Vcf files generated to detect the presence of clonal hematopoiesis are available on the European Variation Archive repository under the identifier PRJEB82347. The bioinformatic process that allowed to generate these data

is fully described in the Method section. Raw sequencing data cannot be shared due to the European General Data Protection Regulation. In addition, the transfer of all or part of the research database must be authorized by the sponsors of the CHAth and Three-city studies, and must be the subject of a written contract. Interested researchers can access the original raw data by contacting the corresponding author and submitting a project proposal to the Research and Innovation Department of the University Hospital of Bordeaux (for the CHAth study, drci@chu-bordeaux.fr) or to 3C's Scientific Advisory Board (for the 3C study, E3C.CoordinatingCenter@u-bordeaux.fr). Commercial research may be carried out on the data under the same conditions.

The following dataset was generated:

| Author(s) | Year | Dataset title | Dataset URL | Database and Identifier |
|---|---|---|---|---|
| Couffinhal T, Mansier O | 2024 | Frequency of clonal haematopoiesis in patients with a first cardiovascular event after 75 years of age: Consequences on inflammation | https://www.ebi.ac.uk/eva/?eva-study=PRJEB82347 | European Variation Archive, PRJEB82347 |

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
