## [Editor Report · eLife assessment]

In this small study involving patients with a history of myocardial infarction, Fawaz et al. found no significant contribution of clonal hematopoiesis and mosaic loss of the Y chromosome to the incidence of myocardial infarction and atherosclerosis. Although the evidence provided by the study is **incomplete** due to its small sample size, the findings are **valuable** for guiding future larger studies that will further investigate this significant and controversial subject.

---

## [Referee Report · Reviewer #2 (Public review)]

Summary:

The preprint by Fawaz et al. presents the findings of a study that aimed to assess the relationship between somatic mutations associated with clonal hematopoiesis (CHIP) and the prevalence of myocardial infarction (MI). The authors conducted targeted DNA sequencing analyses on samples from 149 MI patients and 297 non-MI controls from a separate cohort. Additionally, they investigated the impact of the loss of the Y chromosome (LOY), another somatic mutation frequently observed in clonally expanded blood cells. The results of the study primarily demonstrate no significant associations, as neither CHIP nor LOY were found to be correlated with an increased prevalence of MI. The null findings regarding CHIP are partly in conflict with several larger studies in the literature. However, it must be noted that the authors did find trends to an association between CHIP and a higher incidence of MI during follow-up among those without a history of MI at baseline, which is more consistent with previous research work. The association with incident MI reached statistical significance in men, particularly in those not showing LOY, suggesting potential interactions between different clonally-expanded somatic mutations.

Strengths:

Overall, this is a useful research work on an emerging risk factor for cardiovascular disease (CVD). The use of a targeted sequencing approach is a strength, as it offers higher sensitivity than the whole exome sequencing approaches used in many previous studies. Reporting null findings is definitely relevant in an emerging field such as the role of somatic mutations in cardiovascular disease.

Weaknesses:

The study suffers from important limitations, which cast some doubts onto the authors' conclusions, as detailed below:

(1) The small sample size of the study population is a critical limitation, particularly when reporting null findings that conflict (partly) with positive findings in much larger studies, totaling hundreds of thousands of individuals (e.g. Zekavat et al, Nature CVR 2023, Vlasschaert et al, Circulation 2023; Zhao et al, JAMA Cardio 2024). The authors claim that they have 90% power to detect an effect size of CHIP on MI comparable to that in previous reports (a hazard ratio of 1.7, mainly based on the findings by Jaiswal et al, NEJM 2014,2017). However, this analysis is simply based on the predicted prevalence of CHIP in MI(+) and MI(-) patients, and it does not consider the complex relationship between age CHIP and atherosclerotic disease. More advanced approaches to calculate statistical power may have provided a more accurate estimation. It must also be noted that recent work in much larger populations suggest that the overall effect of CHIP on atherosclerotic CVD is smaller than 1.7, most likely due to the heterogeneity of effects of different mutated genes (e.g. Zekavat et al, Nature CVR 2023, Vlasschaert et al, Circulation 2023; Zhao et al, JAMA Cardio 2024). In addition, several analyses in the current manuscript are conducted separately in MI(+) (n = 149) and MI(-) (N=297) individuals, further limiting statistical power. Power is even lower in the investigation of the effects of LOY and its interaction with CHIP, as only men are included in these analyses. Overall, I believe the study is underpowered from a statistical point of view, so the authors' findings need to be interpreted with caution.

(2) Related to the above, it is widely accepted that the effects of CHIP on CVD are highly heterogeneous, as some mutated genes appear to have a strong impact on atherosclerosis, whereas the effect of others is negligible (e.g. Zekavat et al, Nature CVR 2023, Vlasschaert et al, Circulation 2023, among others). TET2 mutations are frequently considered a "positive control", given the multiple lines of evidence suggesting that these mutations confer a higher risk of atherosclerotic disease. However, no association with MI or related variables was found for TET2 mutations in the current work, which likely reflects the limited statistical power of the study to assess accurately the effects of CHIP mutations on atherosclerotic disease.

(3) One of the most essential features of CHIP is the tight correlation with age. In this study, the effect of age on CHIP (e.g. Supp. Tables S5, S6) is statistically significant, but substantially milder than in previous studies. Given the relatively modest effect size of age on CHIP here, it is not surprising that no association with MI or atherosclerotic disease was found, considering that this association would have a much smaller effect size. It must be considered, however, that the advanced age of the population may have confounded the analysis of these relationships, as acknowledged by the authors.

(4) CHIP represents just one type of clonal hematopoiesis (e.g. see https://doi.org/10.1182/blood.2023022222). In this context, it must be noted that the mutated genes included in the definition of "CHIP" here are markedly different than in most previous studies, particularly when considering specifically the studies that demonstrated an association between CHIP and atherosclerotic CVD. For instance, the definition of CHIP in this manuscript includes genes such as ANKRD26, CALR, CCND2, DDX41... that are not prototypical CHIP genes. This is unlikely to have major impact on the main results, as the vast majority of mutations detected are indeed in bona fide CHIP genes, but it needs to be considered when interpreting the authors' findings. Furthermore, the strategy used here for CHIP variant calling and curation is substantially different than that used in previous studies. This is important, because such differences in the definition of CHIP and the curation of variants are at the basis of most conflicting findings in the literature regarding the effects of this condition. The authors estimate that the effect of these discrepancies on the definition of CHIP is limited, but small differences can have substantial impact in a study with limited sample size.

(5) A major limitation of the current study is the cross-sectional design of most of the analyses. For instance, it is not surprising that no association is found between CHIP and prevalent atherosclerosis burden by ultrasound imaging, considering that many individuals may have developed atherosclerosis years or decades before the expansion of the mutant clones, limiting the possible effect of CHIP on atherosclerosis burden. Similarly, the analysis of the relationship between CHIP and a history of MI may be confounded by the potential effects of MI on the expansion of mutant clones. In this context, it is noteworthy that the only positive results here are found in the analysis of the relationship between CHIP at baseline and incident MI development over follow-up. A larger sample size in these longitudinal analyses would provide deeper insights into the relationship between CHIP and MI.

---

## [Author Response]

The following is the authors’ response to the original reviews.

**Reviewer #1 (Public Review):**
This manuscript examines the individual and dual effects of CHIP and LOY in MI employing a cohort of ~460 individuals. CHIP is assessed by NGS and LOY is assessed by PCR. The threshold for CHIP is set at 2% (an arbitrary cutoff that is often used) and LOY at 9% (according to the Discussion text - this reviewer may have missed the section that describes why this threshold was employed). The investigation assessed whether LOY could modulate inflammation, atherosclerotic burden, or MI risk associated with CHIP. Neither CHIP nor LOY independently affected hsCRP, atherosclerotic burden, or MI incidence, nor did LOY presence diminish these outcomes in CHIP+ male subjects.This study represents the first dual analysis of CHIP and LOY on CVD outcomes. The results are largely negative, contradictory to other studies (many with much larger sample sizes). I would attribute the limitation of sample size as a major contributor to the negative data. While the negative data are suspect, the "positive" finding that LOY abolishes the prognostic significance of CHIP on MI is of interest (and consistent with what is understood from mechanistic studies).Overall, I enjoyed reading the paper, and it is of interest to the research community.However, I disagree with some of the authors' interpretations of the data.Generally, many conclusions on CHIP interpretation are based on the comparison of findings from very large datasets that have been evaluated by shallow NGS DNA sequencing. These studies lack sensitivity and accuracy, but this is counterbalanced by their very large sample sizes. Thus, they draw conclusions from the sickest individuals (ICD codes) with the largest clones (explaining the 10% VAF threshold). Here, the study has a well-phenotyped cohort, but as far as this reviewer can tell, the DNA sequencing is "shallow" NGS. Typically, to assess smaller datasets, investigators employ an error-correction method (DNA barcodes, duplex sequencing, etc.) for the sensitivity and accuracy of calling variants. Thus, the current study appears to suffer from this limitation (small sample sizes combined with NGS).

We thank the reviewer for his/her positive and open comment. We acknowledge that we did not use error-corrected sequencing method for our study. However, we do not fully agree with the statement that our NGS sequencing technique is “shallow”.

Considering our entire sequencing panel, we achieve a sequencing depth ≥100X and ≥300X for 100% [99%;100%] and 99% [99%;100%] of the targeted regions respectively. This corresponds to a median depth of 2111X [1578;2574] for all regions sequenced. When considering “CHIP genes”, the median depth is 2694X [1875;3785] for patients from the CHAth study and 3455X [2266;4885] for patients from the 3C study. More specifically, for *DNMT3A* and *TET2* genes, the median depths of sequencing are 2531X [1818;3313] and 3710X [2444;4901] for patients from the CHAth and 3C study respectively. These values are far much higher than the 300X recommended for NGS sequencing by capture technology by the French National Institute of Cancer. Coupling this high depth of sequencing with our bioinformatic pipeline that uses 3 different variant callers, a manual curing for all variants by trained hematobiologists and a bioinformatic tool to estimate the background noise allow us to detect somatic mutation with a VAF of 1% with a high accuracy. Noteworthy, our accuracy in detecting mutations in leukemia-associated genes is tested twice a year as part of our quality control program organized by the French Group of Molecular Biologists in Hematology (GBMHM). We added the information about the depth of sequencing in the Supplementary Methods section.

While the "negative" data from this study are inconclusive, the positive data (i.e. CHIP being prognostic for MI in the absence but not presence of MI) is of interest. Thus, the investigators may want to consider a shorter report that largely focuses on this finding.

We thank the reviewer for his/her interest in this result. We also agree that it would be interesting to focus specifically on demonstrating the impact of mLOY in countering the cardiovascular risk associated with CHIP. We performed additional analysis to demonstrate that this effect was independent of age and cardiovascular risk factors and included this information in the results section.

However, we believe that it is also of interest to show negative results that, although probably due to limitation in sample size, suggest that the cardiovascular risk associated with CHIP is not as strong and clinically pertinent as initially suggested. Of note, if CHIP really increase the risk of Myocardial Infarction in a significant manner, they would be more frequently detected in subjects who suffered from a MI compared to those who did not, which was not observed in our cohort. Moreover, we were able to determine that if CHIP increases the risk of MI, they do it to a much lesser extent (HR = 1.03 for CHIP) -than other established cardiovascular risk factors such as hypercholesterolemia or tobacco use HR = 1.47 and HR = 1.86 respectively in our cohort, which questions the pertinence of considering for CHIP in the management of patients with atherothrombosis. These data have been added in the Results and Discussion sections.

We also believe that our study has the merit to assess directly the impact of CHIP on atheroma burden, which has been performed in only a limited number of studies in the context of coronary artery disease. This could not be possible by analyzing only male subjects in our cohort because it would further decrease the statistical power of our analyses.

**Reviewer #2 (Public Review):**
Summary:The preprint by Fawaz et al. presents the findings of a study that aimed to assess the relationship between somatic mutations associated with clonal hematopoiesis (CHIP) and the prevalence of myocardial infarction (MI). The authors conducted targeted DNA sequencing analyses on samples from 149 MI patients and 297 non-MI controls from a separate cohort. Additionally, they investigated the impact of the loss of the Y chromosome (LOY), another somatic mutation frequently observed in clonally expanded blood cells. The results of the study primarily demonstrate no significant associations, as neither CHIP nor LOY were found to be correlated with an increased prevalence of MI. Of note, the null findings regarding CHIP are in conflict with several larger studies in the literature.Strengths:Overall, this is a useful research work on an emerging risk factor for cardiovascular disease (CVD). The use of a targeted sequencing approach is a strength, as it offers higher sensitivity than the whole exome sequencing approaches used in many previous studies.Weaknesses:Reporting null findings is definitely relevant in an emerging field such as the role of somatic mutations in cardiovascular disease. Nevertheless, the study suffers from severe limitations, which casts doubts on the authors' conclusions, as detailed below:(1) The small sample size of the study population is a critical limitation, particularly when reporting null findings that conflict (partly) with positive findings in much larger studies, totaling hundreds of thousands of individuals (e.g. Zekavat et al, Nature CVR 2023, Vlasschaert et al, Circulation 2023; Zhao et al, JAMA Cardio 2024). The authors claim that they have 90% power to detect an effect size of CHIP on MI comparable to that in a previous report (Jaiswal et al, NEJM 2017). However, the methodology used to estimate statistical power is not described.

We thank the reviewer for his/her pertinent and constructive comments. We totally agree that our study presents a substantially smaller sample size as compared to the studies of Zekavat *et al*, Vlasschaert *et al* or Zhao *et al*.

The CHAth study was designed as a prospective study (which is not frequent in CHIP reports) to demonstrate that, if CHIP increase the risk of MI, they would be detected more frequently in patients who suffered from a MI compared to those who did not. To achieve this, we defined eligibility criteria to have a rather high prevalence of CHIP and optimize the statistical power of a study based on a limited number of patients. We thus enrolled patients who suffered from a first MI after the age of 75 years. These patients had to be compared with subjects from the Three-City study who had 65 years or more at inclusion and did not present any cardiovascular event before inclusion.

To determine the number of patients necessary to achieve our objective, we considered a CHIP prevalence of 20% in the general population after the age of 75 years, as estimated when we set up our study (Genovese *et al*, NEJM 2014, Jaiswal *et al*, NEJM 2014, Jaiswal *et al*, NEJM 2017). At this time the relative risk of MI associated with CHIP was shown to be 1.7, leading to an expected prevalence of CHIP of 37% in subjects who presented a MI. Based on these hypotheses, the recruitment of 112 patients in the CHAth would have been sufficient to detect a significant higher prevalence of CHIP in MI(+) patients compared to MI(-) subjects with a power of 0.90 at a type I error rate of 5%. These calculations were performed by the Research Methodology Support Unit of the University Hospital of Bordeaux. These data were added in the Supplementary Methods section to expose more clearly the design and objectives of the CHAth study.

Finally, we recruited 149 patients in the CHAth study and compared them to 297 control subjects. Although recruiting more patients than initially needed, we observed a similar prevalence of CHIP between our 2 cohorts, suggesting that the cardiovascular risk associated with CHIP is lower than the 1.7 increased risk claimed in most publications related to CHIP in the cardiovascular field. We have to notice that our study was not designed to demonstrate the impact of CHIP on the occurrence of MI during follow-up, which could explain our negative results due to a limited number of patients as stated by the reviewers. This statement has been added in the Supplementary Methods section. However, performing such analysis allowed us to confirm that the risk of MI associated with CHIP was lower than 1.7 and lower than the one associated with hypercholesterolemia or smoking.

We would like also to notice that the eligibility criteria for both CHAth and the Three-City study can have led to a selection bias, possibly contributing to the contradiction of our results with other studies. As stated before, in the CHAth study, only patients who experience a first MI after the age of 75 were enrolled. In the Three-City study, all subjects had 65 years or more at inclusion. On the contrary, most of the cohorts showing an association between CHIP and cardiovascular events were composed of younger subjects:

- Bioimage : median age 70 years (55-80 years)

- MDC : median age 60 years

- ATVB : subjects with a MI before 45 years

- PROMIS : subjects between 30 and 80 years

- UK Biobank : between 40 and 70 years at inclusion, median age of 58 years in the study of Vlasschaert et al.

- Zhao et al : median age of 53.83 years (45.35-62.39 years).

This last information was added in the Discussion section (lines 452-454).

Furthermore, the work by Jaiswal et al (NEJM 2017) showed a hazard ratio of approx. 2.0, but more recent work in much larger populations suggests that the overall effect of CHIP on atherosclerotic CVD is smaller, most likely due to the heterogeneity of effects of different mutated genes (e.g. Zekavat et al, Nature CVR 2023, Vlasschaert et al, Circulation 2023; Zhao et al, JAMA Cardio 2024).

We thank the reviewer for insisting on the fact that the initial HR of 2.0 observed by Jaiswal *et al* was shown to be smaller in more recent studies. This corresponds to what we wrote in the introduction (lines 103-109) and discussion (lines 365-370, 465-471).

In addition, several analyses in the current manuscript are conducted separately in MI(+) (n = 149) and MI(-) (N=297) individuals, further limiting statistical power. Power is still lower in the investigation of the effects of LOY and its interaction with CHIP, as only men are included in these analyses. Overall, I believe the study is severely underpowered, which calls into question the validity of the reported null findings.

We agree with the reviewer that the statistical power of our study is lower than the one of other studies, in particular those based on several hundred thousand patients. Whenever possible, we analyzed our data by combining MI(+) and MI(-) subjects. However, for some aspects such as atherosclerosis, we did not have the same parameters available for these 2 groups and had to analyze them separately, leading to a more limited statistical power. We also have to acknowledge that our study was not designed to demonstrate an effect of CHIP on incident MI (as stated before), limiting our statistical power to demonstrate an effect of CHIP +/- mLOY on the incident risk of coronary artery disease.

However, when designing our prospective study (CHAth study), we aimed to address the limitations of a small cohort and obtain rapid, significant results regarding the impact of CHIP. We hypothesized that if CHIP really increases the risk of myocardial infarction (MI), it would be detected more frequently in patients who have experienced a MI compared to those who have not. This study design would demonstrate the importance of CHIP in MI pathophysiology without requiring thousands of patients. However, we did not observe such an association questioning the relevance of detecting CHIP for the management of patients in the field of Cardiology. This was confirmed by the fact that in our cohort, the cardiovascular risk associated with CHIP appears to be low (HR = 1.03 [0.657;1.625] after adjustment on sex, age and cardiovascular risk factors) compared to hypercholesterolemia (HR = 1.474 [0.758;2.866]) or smoking (HR = 1.865 [0.943;3.690]). These data have been added in the Results and Discussion sections.

In addition, we would like to mention that despite the limited number of subjects studied, we do not have only negative results. When studying only men subjects, we were able to show that CHIP accelerate the occurrence of MI, particularly in the absence of mLOY (Figure 2D). This effect was independent of age and cardiovascular risk factors (diabetes, cholesterol and high blood pressure). We added this last information in the results section of the manuscript, although we acknowledge that this has to be confirmed in future work.

(2) Related to the above, it is widely accepted that the effects of CHIP on CVD are highly heterogeneous, as some mutated genes appear to have a strong impact on atherosclerosis, whereas the effect of others is negligible (e.g. Zekavat et al, Nature CVR 2023, Vlasschaert et al, Circulation 2023, among others). TET2 mutations are frequently considered a "positive control", given the multiple lines of evidence suggesting that these mutations confer a higher risk of atherosclerotic disease.However, no association with MI or related variables was found for TET2 mutations in the current work. Reporting the statistical power specifically for assessing the effect of TET2 mutations would enhance the interpretation of these results.

We thank the reviewer for this pertinent remark. It has indeed been shown that depending on the somatic mutation, the impact of CHIP on inflammation, atherosclerosis and cardiovascular risk is different. The studies cited by the reviewer suggest that *DNMT3A* mutations have a low impact on atherosclerosis/atherothrombosis while other “non-*DNMT3A*” mutations, including *TET2* mutations, have a greater impact. In particular, Zekavat *et al* suggested that *TP53*, *PPM1D*, *ASXL1* and spliceosome mutations have a similar impact on atherosclerosis/atherothrombosis to *TET2*.

To answer to the reviewer in our cohort, we did not find a clear association between the detection of *TET2* mutation with a VAF≥2% and:

- A history of MI at inclusion (p=0.5339)

- Inflammation (p=0.440)

- Atherosclerosis burden :

- In the CHAth study:

- p=0.031 for stenosis≥50%

- p=0.442 fir multitruncular lesions

- p=0.241 for atheroma volume

- in the 3C study :

- p=0.792 for the presence of atheroma

- p=0.3966 for the number of plaques

- p=0.876 for intima-media thickness

- Incidence of MI (p=0.5993)

Similarly we did not find any association between the detection of *TET2* mutations with a VAF≥1% and:

- A history of MI at inclusion (p=0.5339)

- Inflammation (p=0.802)

- Atherosclerosis burden :

- In the CHAth study :

- p=0.104 for stenosis≥50%

- p=0.617 fir multitruncular lesions

- p=0.391 for atheroma volume

- in the 3c study:

- p=0.3291 for the presence of atheroma

- p=0.2060 for the number of plaques

- p=0.2300 for intima-media thickness

- Incidence of MI (p=0.195)

However, analyzing the specific effect of *TET2* mutations reduces the cohort of CHIP(+) subjects to 61 individuals. In these conditions, considering a prevalence of “*TET2*-CHIP” of 13.5% (in our cohort) and a hazard ratio of 1.3 (Vlasschaert *et al*), the statistical power to show an increased risk of MI is only 16%.

(3) One of the most essential features of CHIP is the tight correlation with age. In this study, the effect of age on CHIP (Supplementary Tables S5, S6) seems substantially milder than in previous studies. Given the relatively weak association with age here, it is not surprising that no association with MI or atherosclerotic disease was found, considering that this association would have a much smaller effect size.

We thank the reviewer for highlighting this point. Although the difference of median age between subjects with or without a CHIP is not very important in our cohort, we did observe a significant association of CHIP with age:

- The differences in age were statistically significant both in the CHAth and 3C study (Supplementary Tables S5 and S6)

- We observed a significant association between age and CHIP prevalence (p<0.001 for the total cohort, p=0.0197 for the CHAth study, and p=0.0394 for the 3C cohort after adjustment on sex). This association was already shown in the figure 1. We added the significant association between age and CHIP prevalence in the Results section (line 279).

As stated before, we have to remind the reviewer that we enrolled only subjects of ≥75 years and ≥65 years in the CHAth and 3C studies respectively. This led to a median age in our cohort that was substantially higher than in other cohorts (in particular the UK Biobank and the different cohorts studied by Jaiswal et al). This could have contributed to an apparent milder effect of age on CHIP, even if this association was still observed.

In addition, there are previous reports of sex-related differences in the prevalence of CHIP, is there an association between CHIP and age after adjusting for sex?

The reviewer correctly pointed out that sex has been associated with various aspects of CHIP. While Zekavat *et al* reported that CHIP carriers were more frequently males, Kar *et al* (Nature *Genetics 2022)*, and Kamphuis *et al* (Hemasphere 2023) did not observe a difference in the prevalence of CHIP between males and females, but rather a difference in the mutational spectrum. Male presented more frequently *SRSF2*, *ASXL1*, *SF3B1*, *U2AF1*, *JAK2*, *TP53* and *PPM1D* mutations while females had more frequently *DNMT3A, CBL* and *GNB1* mutations.

In our study, the association between CHIP prevalence and age was indeed significant even after adjustment on sex (p<0.001 for the total cohort, p=0.0197 for the CHAth study and p=0.0394 for the 3C).

(4) The mutated genes included in the definition of "CHIP" here are markedly different than those in most previous studies, particularly when considering specifically the studies that demonstrated an association between CHIP and atherosclerotic CVD. For instance, the definition of CHIP in this manuscript includes genes such as ANKRD26, CALR, CCND2, and DDX41... that are not prototypical CHIP genes. This is unlikely to have a major impact on the main results, as the vast majority of mutations detected are indeed in bona fide CHIP genes, but it should be at least acknowledged.

We agree with the reviewer that our gene panel includes genes that are not considered prototypical CHIP genes. This acknowledgment has been added in the Supplementary Methods section. To perform this study, we did not design a specific targeted sequencing panel. We used the one that is used for the diagnosis of myeloid malignancies at the University Hospital of Bordeaux. *ANKRD26* and *DDX41* are genes that, when mutated, predispose to the development of hematological malignancies. *CALR* mutations are frequently detected in Myeloproliferative Neoplasms while *CCND2* mutation can be detected in acute myeloid leukemia among other diseases. As usually performed in our routine practice, we analyzed all the genes in the panel. However, as stated by the reviewer, most of the mutations we detected involved *bona fide* CHIP genes.

Furthermore, the strategy used here for the CHIP variant calling and curation seems substantially different than that used in previous studies, which precludes a direct comparison. This is important because such differences in the definition of CHIP and the curation of variants are the basis of most conflicting findings in the literature regarding the effects of this condition. Ideally, the authors should conduct sensitivity analyses restricted to prototypical CHIP genes, using the criteria that have been previously established in the field (e.g. Vlasschaert et al, Blood 2023).

We agree with the reviewer, our strategy for CHIP variant calling and curation was substantially different from what has been used in other studies. We decided to apply the criteria we used in previous studies for the analysis of somatic mutation in myeloid malignancies. Because CHIP are defined by the detection of “somatic mutations in leukemia driver genes”, this appeared to follow the definition of CHIP.

We also acknowledge that this discrepancy with the criteria defined by Vlasschaert *et al* could contribute to our findings that differ from those of other studies. We thus checked whether the variants detected were in accordance or not with the criteria defined by Vlasschaert *et al*. Pooling the 2 cohorts, we detected 439 variants, 381 of which were in accordance with the criteria established by Vlasschaert *et al*, representing a concordance rate of 86.8%. Moreover, the variants “wrongly” retained according to these criteria had an impact on the conclusion on the detection of CHIP in only 15 patients (because these variants were associated with a mutation in a *bona fide* CHIP gene and/or because its VAF was below 2%). Thus, the impact of CHIP variant calling and curation had only a limited impact on our results. This has been added in the discussion (lines 455-459).

However, we would like to discuss the criteria that have been defined by Vlasschaert *et al* which are probably too restrictive. For some genes, such as *ZRSR2*, in addition to frameshift and non-sens mutations that are expected to be associated with a loss of function, only some single nucleotide variations were retained (probably those detected by this group). In our patient 20785, we detected a c.524A>G, p.(Tyr175Cys) mutation that was not reported in the list published by Vlasscheart *et al*. However, this variant presents a VAF presumptive of a somatic origin (3%), affects the Zn finger domain of the protein and is observed in a male subject. Thus, it presents several criteria to consider it as associated with a loss of function. Similarly, the *CBL* variant c.1139T>C, p.(Leu380Pro) observed in our patient 21536, although not affecting the residues 381-421 of the protein (the criteria defined by Vlasschaert *et al*), has been reported in 29 cases of hematological malignancies. It is thus likely to have a significant impact on the behavior of hematopoietic cells. Moreover, in the same patient, a *TET2* c.4534G>A, p.(Ala1512Thr) variant was detected. Although not affecting directly the CD1 domain, it has been reported in a case of AML with a VAF suggestive of a somatic origin (Papaemmanuil *et al*, NEJM 2016). The *SH2B3* gene is not considered by Vlasschaert *et al* as a *bona fide* CHIP gene, contrary to other genes involved in cell signaling such as *JAK2*, *GNAS*, *GNB1*, *CBL*. However, inactivating mutations in *SH2B3* can be detected in myeloid malignancies and were recently shown to drive the phenotype in some patients with a MPN (Zhang *et al*, American Journal of Hematology 2024). We could thus expect that this also happens in our patients 22591 and 21998 who harbor mutations of *SH2B3* (a SNV in the PH domain and a frameshift mutation respectively).

Regarding *BCOR*, *STAG2*, *SMC3* and *RAD21* genes, although frameshift mutations are the most prevalent, there are several reports on the existence of SNV in the context of hematological malignancies (COSMIC, Blood (2021) 138 (24): 2455–2468, Blood Cancer Journal (2023)13:18 ; https://doi.org/10.1038/s41408-023-00790-1).

We can also add that although Vlasschaert *et al* did not consider *CSF3R* and *CALR* as CHIP-genes, Kessler *et al* did. Because CHIP are an emerging field, it should be considered that the concepts that define it are expected to evolve, as demonstrated by the recent study of the Jyoti Nangalia’s group (Bernstein et al, Nature Genetics 2024) who showed that 17 additional genes (including *SH2B3*) should be considered as driver of clonal hematopoiesis.

(5) An important limitation of the current study is the cross-sectional design of most of the analyses. For instance, it is not surprising that no association is found between CHIP and prevalent atherosclerosis burden by ultrasound imaging, considering that many individuals may have developed atherosclerosis years or decades before the expansion of the mutant clones, limiting the possible effect of CHIP on atherosclerosis burden. Similarly, the analysis of the relationship between CHIP and a history of MI may be confounded by the potential effects of MI on the expansion of mutant clones. In this context, it is noteworthy that the only positive results here are found in the analysis of the relationship between CHIP at baseline and incident MI development over follow-up. Increasing the sample size for these longitudinal analyses would provide deeper insights into the relationship between CHIP and MI.

We agree with the reviewer that increasing the sample size for longitudinal analyses would provide deeper insights into the relationship between CHIP and MI. Unfortunately, for the moment, we do not have access to additional samples of the 3C study and are not able to perform these additional analyses.

(6) The description of some analyses lacks detail, but it seems that statistical analyses were exclusively adjusted for age or age and sex. The lack of adjustment for conventional cardiovascular risk factors in statistical analyses may confound results, particularly given the marked differences in several variables observed between groups.

The reviewer is right when saying that we adjusted our analyses on age and/or sex. This was done because as stated before, our results did not show a lot of significant differences. However, we reanalyzed our data, adjusting further the tests for conventional cardiovascular risk factors, and observed similar results. These data have been added in the results section (lines 286-287, 303, 319, 331-332, 341).

(7) The variant allele fraction (VAF) threshold for identifying clinically relevant clonal hematopoiesis is still a subject of debate. The authors state that subjects without any detectable mutation or with mutations with a VAF below 2% were considered non-CHIP carriers. While this approach is frequent in the field, it likely misses many impactful mutations with lower VAFs. Such false negatives could contribute to the null findings reported here. Ideally, the authors should determine the lower detection limit of their sequencing approach (either computationally or through serial dilution experiments) and identify the threshold of VAF that can be detected reliably with their sequencing assay. The association between CHIP and MI should then be evaluated considering all mutations above this VAF threshold, in addition to sensitivity analyses with other thresholds frequent in the literature, such as 1% VAF, 2% VAF, and 10% VAF.

We agree with the reviewer that the VAF threshold for identifying clinically relevant CH is still debated. As stated in the manuscript and by the reviewer, we used the conventional threshold of 2%. Considering that different studies have shown that the cardiovascular risk is increased in a more important manner for CHIP with a high VAF (Jaiswal et al, NEJM 2017, Kessler et al Nature 2022, Vlasschaert et al, Circulation 2023), it is not sure that considering variant with a very low VAF (below 2%) would help us in finding an impact of CHIP on inflammation, atherosclerosis or atherothrombotic risk.

However, as mentioned by the reviewer, variants with a low VAF could have a clinical impact as recently reported by Zhao *et al*. In France, the use of biological analysis for medical purposes imposes to demonstrate that all its aspects are mastered, including their performances. In that context, we determined that our NGS strategy allowed us to reliably detect mutation with a VAF down to 1% (data not shown). As stated in the discussion, we also analyzed our results considering variants with a VAF of 1% and found similar results (lines 394-395). The sensitivity analyses were already mentioned in the manuscript, as we also searched for an effect of CHIP with a high VAF (≥5%) and found no effect neither. We did not have a sufficient number of subjects carrying variants with a VAF≥10% to perform analysis with this threshold.

(8) The authors should justify the use of 3D vascular ultrasound imaging exclusively in the supra-aortic trunk. I am not familiar with this technique, but it seems to be most typically used to evaluate atherosclerosis burden in superficial vascular beds such as carotids or femorals. I am concerned about the potential impact of tissue depth on the accurate quantification of atherosclerosis burden in the current study (e.g. https://doi.org/10.1016/j.atherosclerosis.2016.03.002). It is unclear whether the carotids or femorals were imaged in the study population.

We apologize for the lack of precision in the Methods section. As stated by the reviewer, we evaluated the atherosclerosis burden in superficial vascular beds. We measured atheroma volume at the site of the common carotid (as described by B Lopez-Melgar, in Atheroslerosis, 2016). We did not analyze femoral arteries in this study. The sentence is now corrected in the Methods (lines 176-179).

(9) The specific criteria used to define LOY need to be justified. LOY is stated to be defined based on a "A cut off of 9% of cells with mLOY defined the detection of a mLOY based on the study of 30 men of less than 40 years who had a normal karyotype as assessed by conventional cytogenetic study." As acknowledged by the authors, this definition of LOY is substantially different than that used in recent studies employing the same technique to detect LOY (Mas-Peiro et al, EHJ 2023). In addition, it seems essential to provide more detailed information on the ddPCR assay used to determine LOY, including the operating range and, more importantly, the lower limit of detection (%LOY) of the assay. A dilution series of a control DNA with no LOY would be helpful in this context.

We apologize if the definition of the threshold for detecting mLOY was unclear. To test the performance of our ddPCR technique, we first determined the background noise by testing DNA obtained from total leukocytes in 30 men of ≤40 years who presented a normal karyotype as assessed by conventional cytogenetic technics. In this control population supposed not to carry mLOY, we detected of proportion of cells with mLOY of 2,34+/-1,98 (see Author response image 1, panel A). We thus considered a threshold above 9% as being different from background noise (mean + 3 times the standard deviation).

We then compared the proportion of cells with mLOY measured by ddPCR and conventional karyotype and observed a rather good correlation between the 2 technics (R2=0.6430, p=0.0053, see Author response image 1, panel B). Finally, we tested the reliability of our ddPCR assay in detecting different levels of mLOY using a dilution series of control DNA (from an equivalent of 2% of cell with mLOY to 98% of cells with mLOY). We observed a very nice correlation between the theoretical and measured proportions of cells with mLOY (R2=0.9989, p<0.001, see Author response image 1, panel C). Of note, the proportion of mLOY measured for values ≤10% were concordant with theoretical values. However, considering the background noise determined with control DNA, we were unable to confirm that this “signal” was different from the background noise. Therefore, we set a threshold of 9% to define the detection of mLOY by ddPCR. It is also noteworthy that the 10% cell population with mLOY was consistently detected by the ddPCR technique. This has been added in the Methods section (lines 228-235).

**Author response image 1. sa2fig1:** 

(10) Our understanding of the relationship between CHIP and CVD is evolving fast, and the manuscript should be considered in the context of recent literature in the field. For instance, the recent work by Zhao et al (JAMA Cardio 2024, doi:10.1001/jamacardio.2023.5095) should be considered, as it used a similar targeted DNA sequencing approach as the one used here, but found a clear association between CHIP and coronary heart disease (in a population of 6181 individuals).

We thank the reviewer for this pertinent reference. We did not include it in the first version of our manuscript because it was not published yet when we submitted our work. We included this reference in the discussion (lines 451, 455, 464). We also included the recent study of Heimlich *et al* (Circ Gen Pre Med 2024, lines 464-468) who studied the association of CHIP with atherosclerosis burden.

(11) The use of subjective terms like "comprehensive" or "thorough" in the title of the manuscript does not align with the objective nature of scientific reporting.

We removed the terms “comprehensive” and “thorough” from the title and the text.

Recommendations for the authors:

**Reviewing Editor:**
The Editors believe that in light of the small study the word Comprehensive has to be removed (including from the title and abstract).

We agree and removed the term comprehensive from the title and the text.

**Reviewer #1 (Recommendations For The Authors):**
Other comments:It has long been recognized that hsCRP does not adequately address the inflammation associated with CHIP. For example, see Bick et al Nature 2020; 586:763. Through an assessment of a large dataset, the regulation of multiple inflammatory mediators was associated with CHIP but not with CRP.

We agree that hsCRP is probably not the most sensitive marker for inflammatory state associated with CHIP. However, it is the most commonly used one in medical practise. However, as indicated in the discussion (lines 418-420), we did not observe any association between CHIP and the plasmatic level of different cytokines (IL1ß, IL6, IL18 and TNFα) in patients enrolled in the CHAth study.

Many of the citations lack journal names, volumes, page numbers, etc.

We apologize for this and corrected the citations.

Please provide more details on the methodology (i.e. is CHIP assessed only through NGS with no error correction?). Specify the rationale for why the 9% LOY threshold was employed. Provide this information in the Methods section.

We added more details on the methodology as demanded in the results section (lines 212-214 and 228-235).

Supplementary Table S3 lacks headings. What are the designations for columns 6-8?

We apologize for this and corrected the Table. Columns 6-8 correspond to the VAF, coverage of the variants and depth of sequencing, as for Table S4.